# Quality-of-Life Assessment of Women Undergoing In Vitro Fertilization in Kazakhstan

**DOI:** 10.3390/ijerph192013568

**Published:** 2022-10-20

**Authors:** Meruyert Suleimenova, Vyacheslav Lokshin, Natalya Glushkova, Sholpan Karibayeva, Milan Terzic

**Affiliations:** 1Department of Public Health and Social Sciences, Kazakhstan Medical University “KSPH”, Almaty 050000, Kazakhstan; 2Department of Assisted Reproductive Technologies, International Clinical Centre of Reproduction “PERSONA”, Almaty 050000, Kazakhstan; 3Department of Epidemiology, Biostatistics and Evidence Based Medicine, Al-Farabi Kazakh National University, Almaty 050000, Kazakhstan; 4Department of Medicine, School of Medicine, Nazarbayev University, Nur-Sultan 010000, Kazakhstan; 5Clinical Academic Department of Women’s Health, Corporate Fund “University Medical Center”, Nur-Sultan 010000, Kazakhstan; 6Department of Obstetrics, Gynecology and Reproductive Sciences, University of Pittsburgh School of Medicine, Pittsburgh, PA 15213, USA

**Keywords:** FertiQoL, infertility, in vitro fertilization, quality of life, socio-demographic factors

## Abstract

Infertility is a problem that affects millions of couples worldwide and has a significant impact on their quality of life. The recently introduced “Fertility Quality of Life Questionnaire (FertiQoL)” quickly became a gold standard for evaluation of the quality of life of patients suffering from infertility. The aim of this study was to determine the quality of life of Kazakhstani women coping with infertility problems by FertiQoL and assess the validity of the questionnaire. This cross-sectional study involved women of reproductive age undergoing an in vitro fertilization (IVF) cycle at a large IVF center in Kazakhstan in the period from 1 September 2020 to 31 September 2021. A total of 453 women out of 500 agreed to participate in the study, and the response rate was 90.6%. The overall Core FertiQoL was 56.95 ± 14.05, and the Treatment FertiQoL was 66.18 ± 11.13 points. Respondents with secondary infertility had statistically significantly higher Emotional (*p* < 0.001), Mind–body (*p* = 0.03), Social (*p* < 0.001), Environment (*p* = 0.02), and Treatment (*p* < 0.001) domains of FertiQoL than women with primary infertility. Respondents with a low income had the lowest levels of Total FertiQoL (56.72 ± 11.65). The longer duration of infertility of women undergoing IVF treatment presented the worse scale of Treatment and Total FertiQoL. Cronbach’s alpha revealed good internal reliability for all FertiQoL subscales on the Kazakhstan women’s questionnaire and averaged 0.8, which is an indicator of a high degree of reliability. The Total FertiQoL of Kazakhstan women undergoing IVF treatment was 59.6 ± 11.5, which is considerably lower than European countries. We identified statistically significant differences across medical and demographic groups. As this questionnaire had validity in Kazakhstan survey it possibly be used for both medical counseling and future investigation in our country.

## 1. Introduction

Parenthood is undoubtedly one of the most desirable goals in adult life, but unfortunately, not all couples can achieve it spontaneously. Infertility is defined as an inability to conceive within twelve months of unprotected, regular intercourse or therapeutic donor insemination of women younger than 35 years or within 6 months in women who are older than 35 years [1]. The World Health Organization has recognized infertility as a public health problem worldwide. Infertility affects about 9% of reproductive age couples, and only half of them seek medical help [2]. According to various sources, the frequency of the problem in Kazakhstan ranges from 12 to 15.5% [3]. Infertility and its treatment significantly affect the quality of a person’s life. Studies show that infertility problems are among the saddest events in people’s lives [4]. It is reported that women undergoing IVF procedures are experiencing various types of psychological burdens such as stress, anxiety and depression [5,6,7].

Quality of life includes areas such as emotional well-being, social functioning, physical health, the patient’s environment, and personal beliefs. Previously, general self-assessment instruments were used to assess the quality of life in infertile patients. However, many of the existing questionnaires on infertility distress and treatment response do not meet fertility-specific requirements [8]. Therefore, a special tool for measuring the quality of life, designed for infertile couples, was developed ten years ago and is used internationally—Fertility Quality of Life (FertiQoL) [9].

The FertiQoL questionnaire is a multidimensional tool designed to assess the quality of life of people with infertility and has quickly become the gold standard for a specific quality-of-life measurement in theoretical and practical questions of infertility [8]. It is specifically designed to assess the quality of life of infertile patients by experts from the European Society for Human Reproduction and Embryology (ESHRE) and the American Society for Reproductive Medicine (ASRM). FertiQoL has been used across cultures and communities and has been translated into 26 languages. FertiQoL has been validated in various countries (see www.fertiqol.org, accessed on 3 September 2022) and has shown good overall psychometric performance [10]. Its usefulness has been confirmed in Dutch, Italian, Iranian and German studies comparing the FertiQoL instrument with other universal QoL measuring instruments [11,12,13], and also demonstrated convincing convergent validity with scales of depression, anxiety, and relationships [14].

In Kazakhstan, there were studies assessing the relationship between age, psychological distress and IVF outcome [5,6,15,16], as well as the impact of governmental support to the IVF clinical pregnancy rates [17], while there were no investigations evaluating the quality of life in women undergoing IVF using fertility-specific measurement. The mentality of the country is based on putting pressure on couples that they need to become parents. In the process of examination and treatment, couples face financial problems and psychological burnout [12].

The aim of this study was to determine the quality of life of Kazakhstani women coping with infertility problems and to validate the questionnaire, FertiQoL.

## 2. Materials and Methods

### 2.1. Patients and Enrollment

This cross-sectional study was conducted at the large International Clinical Center of Reproduction “Persona”, where women from all regions of Kazakhstan are treated for infertility using assisted reproductive technologies. FertiQoL questionnaires were presented to infertile couples who were treated with ART from 1 September 2020 to 31 September 2021. All participants who completed the survey participated in this study voluntarily and anonymously. This study had the following inclusion criteria: (1) Women aged 18–49; (2) Absence of pregnancy after 12 months of regular unprotected sex; (3) Undergoing treatment in clinics of assisted reproductive technologies (ART); (4) Knowledge of the Russian language in speech and writing. Exclusion criteria: (1) Confirmed mental disorders; (2) The presence of background somatic pathology (diabetes, hypertension).

The survey was carried out using the Google Form platform and was sent to patients by the attending physician in ICCR “Persona”. The questionnaire consisted of FertiQoL and questions of additional factors such as the presence of children, financial well-being, infertility duration, place of residence, education and the cause of infertility.

### 2.2. Questionnaire: Fertility Quality-of-Life (FertiQoL) Tool

The FertiQoL tool consists of two main modules: the main FertiQoL module and the additional processing module. The Core FertiQoL module has 24 elements and the Treatment FertiQoL module has 10 elements. Core FertiQoL’s 24 points are subdivided into four domains, including emotional, and physical (mind/body), relationships, and social. The emotional domain assesses the effect of infertility on emotions such as sadness, resentment, or grief. The Mind/body domain refers to the effects of infertility on physical health, cognition, and behavior. The relationship domain and the social domain are used to quantify the impact of infertility on partnerships and social dimensions (e.g., social inclusion, expectations, and support), respectively. The add-on treatment module consists of two domains that are used to assess the surrounding staff and the tolerability of fertility treatments. Items from these domains are randomly presented on the questionnaire and rated on a scale from 0 to 4, where higher scores indicated more favorable quality of life.

The FertiQoL subscale and total scores are calculated and converted to achieve a range of 0 to 100, where higher scores indicate better quality of life [9]. In our study, the Russian version of the FertiQoL questionnaire was used as a tool for measuring the quality of life of Kazakhstan’s infertile couples. We used the Russian language as a high degree of Russian language competence 93.3% in the RK [18], because RK has been part of the USSR for more than seventy years.

### 2.3. Ethics

The study was approved by the Ethics Committee of the Kazakhstan Medical University “Higher School of Public Health” (Protocol No: IRB-A108 dated 19 December 2019).

### 2.4. Statistical Processing

Questionnaire results data were entered into SPSS version 26.0 for aggregation and statistical analysis. For continuous numbers we used a descriptive analysis with an estimation of mean (Me) and standard deviation (SD), as well as absolute numbers (*n*) and percentages (%) for qualitative variables.

Women are divided into two groups according to presence of children: primary infertility, for persons who had no child, and secondary infertility, for women with one or more child [19].

The respondents’ income levels were divided into four quartiles: quartile 1 (under USD 218), quartile 2 (USD 218–545.72), quartile 3 (USD 545.72–982), and quartile 4 above USD 982. The conversion rate was, respectively: USD 1 dollar is equal to KZT 458.7 on 27 February 2022 (converted by https://www.xe.com/currencyconverter/ (accessed on 3 September 2022) [20].

The quality-of-life variable is quantitative, so to identify differences between the means of two groups, two-sample Student’s *t*-test (presence of children, place of residence) were used. One-way analysis of variance (ANOVA) was used to test for differences between means of three or more independent groups, infertility duration, education, family income, the cause of infertility, etc.). A *p*-value ≤ 0.05 was considered statistically significant. Cronbach’s coefficients α have been calculated to assess the reliability of the FertiQoL instrument. Cronbach’s alpha > 0.7 was considered satisfactory [9].

We used data from the website https://www.statista.com (accessed on 7 September 2022) [21] to obtain information on gross domestic product (GDP) and % of GDP for healthcare to link it with women’s quality of life in 5 countries [22,23].

## 3. Results

### 3.1. Characteristics of Participants

In this cross-sectional study, 500 patients who were undergoing IVF treatment were invited to participate; 453 agreed, and the response rate was 90.6%.

The age of the respondent’s ranged from 20 to 49 years (mean age is 34.76 ± 5.89 years). Most of the respondents had completed higher education and lived in urban areas (78.3%). Half of the respondents had an income between USD 218 and USD 545.72, which is below average. The most prevalent duration of infertility among our respondents was 2–5 years (43%) (Table 1).

The Total FertiQoL and its subscale scores are presented in Table 2. The mean Core FertiQoL and Treatment FertiQoL were 56.95 ± 14.05 and 66.18 ± 11.13, respectively. Cronbach’s α coefficients of the FertiQoL subscale averaged 0.8 during validation of questionnaire for Kazakhstani women. All scales of FertiQoL had acceptable internal consistency, and there was no need to delete any items.

We examined the satisfaction scores by FertiQoL, dividing them into two groups according to the presence of children (Table 3). FertiQoL scores were statistically significantly higher in the group with children than in the group with primary infertility in all domains (*p* < 0.05), except for Relation and Tolerability (*p* = 0.60, *p* = 0.48).

When comparing the scores from respondents from rural areas with those from urban areas, we found that scores for the emotional scale were more than two times lower in respondents from rural areas (27.42 ± 9.59 and 63.86 ± 14.99; *p* < 0.001). Rural residents had significantly lower scores than urban women in all FertiQoL scales (*p* < 0.001), except environment item (t = −0.859; D.f. = 157.8; *p* = 0.4) (Table 4).

When analyzing differences in groups divided by income, Q4 (above USD 982) and Q3 (USD 545.72–982) had significantly higher scale in Emotional, Mind/body, Social, Core and Total FertiQoL items than Q1 (up to USD 218), Q2 (USD 218–545.72) (*p* ˂ 0.001) (Table 5).

When comparing respondents with a duration of infertility up to two years with those who had longer period of infertility, the first one showed higher levels of quality of life. We found a statistically significant decrease in Treatment FertiQoL and Total FertiQoL when duration of infertility was longer (*p* = 0.02, *p* = 0.03, respectively) (Table 6).

There is no difference when comparing FertiQoL scores in groups divided by education level (Table 7).

### 3.2. Health Expenditure and Quality of Life

Quality of life of the population also depends on the overall health; therefore, we decided to link the % of GDP for healthcare and Total FertiQoL score.

When comparing Kazakhstan with Turkey, Poland, Israel and Germany, Kazakhstan had the lowest rates of health expenditure and Total FertiQoL, listed on Figure 1.

## 4. Discussion

In our study, we evaluated the general and subscale level of quality of life of women undergoing IVF. We tried to assess what factors may have an impact on the quality of life of Kazakhstani women undergoing infertility treatment. The most of our respondents lived in urban areas (78.3%) and had primary infertility (64.4%).

Many European (German, Italian, French, Poland) and Asian (India, Taiwan, Iran) countries have examined the FertiQoL questionnaire in different groups and with other questionnaires, which allowed us to compare our results with the results of other studies [11,23,24,25,26,27]. We determined the differences in scales of FertiQoL depending on infertility treatment and various characteristics.

In our research, Cronbach’s alpha was more than 0.8 and revealed a good internal reliability of Kazakhstan’s questionnaire, and there was no need to exclude questions. In a study by Volpini et al., 2020, to improve the internal consistency of the scale, the authors especially removed Q4 and T2 items (ranging from 0.91 to 0.70), while in a Taiwanese study, Q11, Q13 and T5 items were deleted (ranging from 0.5 to 0.86) [13,25]. In Dural et al., 2016, as in our study, Cronbach’s alpha was between 0.70 and 0.89, and there was no need to remove questions [28].

In contrast to the study by Donarelli et al., our study showed that women who are already raising a child reported statically significant higher level of quality of life in the Emotional, Mind/body, Social, Environment, Core FertiQoL, Treatment FertiQoL, Total FertiQoL domains, as opposed to women with no children (Table 3) [14]. Despite the fact that Kazakhstan is a secular country, a woman’s social position is highly dependent on her maternal status or the possibility of getting pregnant. This was also showed by the Social subscales of primary and secondary infertile women (57.50 ± 15.78 and 65.42 ± 14.39, respectively, *p* < 0.001) as an indicator of social pressure to women. From the other side, according to the report of Sexty et al., there was no difference in the social subscale in the German population (*p* = 0.032) [11]. In the Iranian study by Hekmatzadeh et al., 2018, there was also evidence of a better quality of life for women with children [29]. In a study by Biovin et al., Core FertiQoL was significantly lower (*p* < 0.001) for participants without children (53.3 ± 16.3) than participants with children (59.5 ± 17.7) [8].

We also measured the differences in quality of life by place of residence and reported that respondents living in rural areas reported lower scores on all FertiQoL domains except for the Environment domain. This finding was consistent with the findings of studies in Taiwan, Germany and Poland [12,23,25].

There was a significant tendency to lower satisfaction on the Core FertiQoL scales and Total FertiQoL, depending on the decrease in income, which may indicate difficulties with coping with infertility treatment or social status (Table 4). A comparison by income level was also conducted in a study from Iran, which reported data consistent with ours [27,30].

In our study, a longer duration of infertility was related to lower scores on Treatment and Total FertiQoL domains (*p* < 0.05). The study results of Karabulut and colleagues, 2016, are consistent with ours, linking long duration of infertility with lower indicators in Mind/body, Social, Tolerability and the Total FertiQoL domains (*p* < 0.05) [31]. In a previous study by Huppelschoten et al., respondents with a short duration of infertility were more likely to report better quality of life [32].

S. Madero et al., 2017, concluded that the socio-cultural background of patients plays a role in shaping their experience of treatment in terms of well-being [26]. Health expenditure can be measured by a combination of objective health outcome indicators, such as life expectancy, healthy life years or self-perceived health [33]. In our study, we tried to compare five countries by % of GDP for healthcare and Total FertiQoL scores. We have noticed a downward trend in the overall quality of life in relation to the decline in % GDP for healthcare. The average indicators of FertiQoL domains of women with infertility in Kazakhstan are significantly lower than in other countries. It was found that one in every seven couples with infertility are getting divorced, taking into account these data, the government of Kazakhstan launched the program “Ansagan Sabi”. Since 2020, this program each year subsidized 7000 in vitro fertilization treatments [34]. However, not everyone can rely on this support. Strict criteria for obtaining subsidy by health insurance for infertility force many couples to take out loans for IVF treatment.

This study has some limitations. In our study only women were interviewed, since women come to IVF clinics on their own, men are more closed to the issue of infertility treatment. Additionally, the cross-sectional design limited the ability to establish causal relationships between infertility and quality of life. To develop a tool with axial sectional design, a follow-up study is needed to develop and apply it to verify its effectiveness. The methodology of our study could be strengthened in the future investigations by physical, psychological, and relational factors for each stage of IVF treatment, as was performed in the study of Volpini et al., 2020. Alongside the FertiQoL questionnaire, they studied quality-of-life measurements in women experiencing infertility at different stages of ART treatment, such as diagnostic, stimulation and transfer [13].

Nevertheless, this is the first study which provides an overview of the quality of life of women with infertility in the Republic of Kazakhstan using the fertility-specific quality-of-life assessment tool FertiQoL. The large sample size and high response rate in this study made it possible to describe in detail the quality of life of participants associated with infertility and the impact of additional aspects on it. The results of the present study demonstrate a difficult life situation in women with infertility. Future research to identify various factors related to the quality of life in infertility using a similar approach will help develop a thorough approach to clinical practice as well as social support from the state. Since this study is the first report on the use of the FertiQoL tool in Commonwealth of Independent States (CIS), it can be used as a guideline for monitoring the impact of infertility on the psycho-emotional assessment among the general population of the country. Further collaborative studies are needed in all CIS countries to determine the impact of the factors already assessed in this study.

These data will provide an opportunity for specialists to understand how important it is to work with the psycho-emotional background of women undergoing infertility treatment. In order to maintain the quality of life of infertility women and to improve their compliance with the medical treatment of these, it is important to address and discuss all identified psychosocial issues and burdens.

## 5. Conclusions

Our study provides an overview of the quality of life in infertile women undergoing in vitro fertilization in Kazakhstan using the infertility-specific quality-of-life assessment tool, FertiQoL. The Total FertiQoL of Kazakhstan women undergoing IVF treatment was 59.6 ± 11.5, which is considerably lower than scores reported in European countries. The FertiQoL questionnaire has good validity and can be used in practical healthcare in our country.

## Figures and Tables

**Figure 1 ijerph-19-13568-f001:**
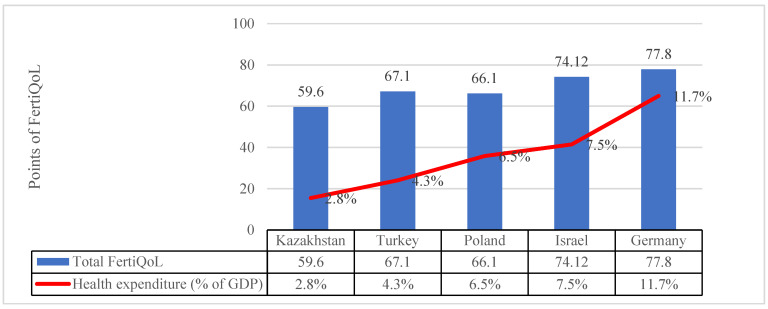
Average satisfaction by Total FertiQoL subscale and health expenditure in % in other countries from 2019 [22,23]. The data was obtained from the website: https://www.statista.com/ (accessed on 7 September 2022) [21].

**Table 1 ijerph-19-13568-t001:** Socio-demographic and clinical characteristics of participants.

Variable	*n* (%)
Average age (in years) *	34.76 (±5.89)
**Infertility type**	
Primary infertility	292 (64.4%)
Secondary infertility	161 (35.6%)
**Infertility duration**	
>1 year	46 (10.2%)
2–5 years	195 (43%)
6–10 years	133 (29.4%)
Over 10 years	76 (16.8%)
**The average level of education**	
Higher education	336 (74.2%)
Incomplete higher education	16 (3.5%)
Specialized secondary	62 (13.7%)
Secondary education	39 (8.6%)
**Place of residence**	
Urban	353 (78.3%)
Rural	98 (21.7%)
**Income**	
Q1 (Low)	100 (22.1%)
Q2 (Below average)	231 (51%)
Q3 (Average)	95 (21%)
Q4 (High)	25 (5.5%)

*—mean ± standard deviation.

**Table 2 ijerph-19-13568-t002:** FertiQoL scores and α-Cronbach’s coefficient.

Subscales of FertiQoL	Mean	SD	Cronbach α
Emotional	51.30	21.05	0.851
Mind/body	57.73	20.76	0.863
Relational	57.28	9.84	0.866
Social	61.48	17.30	0.852
Environment	62.12	11.91	0.873
Tolerability	71.78	18.88	0.861
Core FertiQoL	56.95	14.05	0.833
Treatment FertiQoL	66.18	11.13	0.847

**Table 3 ijerph-19-13568-t003:** Characteristics of average indicators of FertiQoL in groups with primary and secondary infertility.

Items (Mean ± SD)	Primary Infertility (*N* = 292)	Secondary Infertility (*N* = 161)	*t* Test	*p* Value
Total FertiQoL	58.62 ± 11.13	63.05 ± 11.58	−3.675	<0.001
Emotional	54.17 ± 19.72	61.29 ± 21.41	−3.286	0.001
Mind/body	55.36 ± 20.51	60.49 ± 21.62	−2.303	0.02
Relational	56.50 ± 9.97	55.95 ± 10.23	0.516	0.60
Social	57.50 ± 15.78	65.42 ± 14.39	−4.838	<0.001
Environment	60.83 ± 13.55	65.32 ± 10.82	−3.258	0.001
Tolerability	71.48 ± 18.46	72.90 ± 18.65	−0.706	0.48
Core FertiQoL	55.97 ± 13.61	60.76 ± 14.00	−3.267	0.001
Treatment FertiQoL	65.23 ± 12.23	68.72 ± 11.21	−2.708	0.007

**Table 4 ijerph-19-13568-t004:** Characteristics of average indicators of FertiQoL by place of residence.

Items (Mean ± SD)	Urban (*N* = 353)	Rural (*N* = 98)	*t* Test	*p* Value
Total FertiQoL	64.12 ± 9.11	46.94 ± 8.33	−17.691	<0.001
Emotional	63.86 ± 14.99	27.42 ± 9.59	−29.034	<0.001
Mind/body	64.34 ± 18.41	38.21 ± 18.90	−12.177	<0.001
Relational	57.83 ± 10.26	52.54 ± 10.49	−4.413	<0.001
Social	64.75 ± 13.81	45.52 ± 14.34	−11.837	<0.001
Environment	62.66 ± 12.47	61.45 ± 12.06	−0.859	0.4
Tolerability	74.84 ± 17.20	63.48 ± 20.87	−4.881	<0.001
Core FertiQoL	62.66 ± 10.89	40.73 ± 9.00	−20.339	<0.001
Treatment FertiQoL	67.78 ± 11.28	62.29 ± 12.12	−3.977	<0.001

**Table 5 ijerph-19-13568-t005:** Characteristics of average indicators of FertiQoL by income.

Items (Mean ± SD)	Q1 Up to USD 218	Q2 USD 218–545.72	Q3 USD 545.72–982	Q4 Above USD 982	ANOVA (F)	*p* Value
Total FertiQoL	56.72 ± 11.65	60.29 ± 10.69	63.33 ± 11.79	65.78 ± 12.28	8.2	<0.001
Emotional	46.45 ± 20.35	56.5 ± 18.68	61.43 ± 20.16	68.08 ± 22.60	14.5	<0.001
Mind/body	52.69 ± 21.89	58.02 ± 20.23	63.46 ± 20.92	70.54 ± 24.02	7.6	<0.001
Relational	56.39 ± 11.28	56.40 ± 10.30	57.37 ± 10.69	57.40 ± 8.65	0.3	0.851
Social	55.40 ± 16.34	60.23 ± 15.59	65.03 ± 15.20	66.60 ± 14.99	8.1	<0.001
Environment	62.40 ± 11.56	62.62 ± 12.19	62.59 ± 14.63	61.17 ± 8.98	0.1	0.958
Tolerability	74.98 ± 18.96	72.06 ± 17.57	70.59 ± 19.46	71.17 ± 22.74	1.0	0.373
Core FertiQoL	52.59 ± 13.96	57.72 ± 12.86	62.09 ± 14.00	65.69 ± 14.69	11.9	<0.001
Treatment FertiQoL	67.54 ± 11.79	66.66 ± 10.86	66.02 ± 13.66	65.3 ± 10.76	0.4	0.753

**Table 6 ijerph-19-13568-t006:** Characteristics of average indicators of FertiQoL by duration of infertility.

Items (Mean ± SD)	1–2 y	2–5 y	6–10 y	≥10 y	ANOVA (F)	*p* Value
Total FertiQoL	64.57 ± 10.78	54.45 ± 10.53	54.61 ± 10.11	52.60 ± 10.58	2.97	0.03
Emotional	62.59 ± 18.69	49.85 ± 19.51	49.19 ± 19.84	48.25 ± 21.27	1.90	0.13
Mind/body	60.98 ± 11.18	53.87 ± 18.49	54.42 ± 18.37	53.84 ± 21.10	2.45	0.06
Relational	71.20 ± 15.59	56.52 ± 11.23	56.64 ± 11.70	56.52 ± 9.51	0.11	0.95
Social	55.80 ± 12.98	57.88 ± 15.64	57.21 ± 15.18	57.68 ± 13.32	0.78	0.51
Environment	62.88 ± 14.38	54.42 ± 13.28	55.05 ± 13.39	54.07 ± 13.43	2.06	0.11
Tolerability	69.61 ± 19.48	56.73 ± 15.30	56.71 ± 12.83	50.82 ± 13.54	2.55	0.06
Core FertiQoL	63.68 ± 14.43	47.46 ± 16.28	47.60 ± 15.23	42.41 ± 16.97	2.11	0.10
Treatment FertiQoL	65.01 ± 10.13	52.02 ± 14.21	51.86 ± 12.48	45.97 ± 15.41	3.18	0.02

**Table 7 ijerph-19-13568-t007:** The relationship between FertiQoL and the education levels.

Educational Level	The Average	Specialized Secondary	Incomplete Higher Education	Higher Education	F	*p*
Emotional	46.69 ± 19.19	50.74 ± 19.42	50.78 ± 22.01	51.14 ± 20.44	1.2	0.3
Mind/body	53.85 ± 15.95	56.38 ± 19.17	49.22 ± 21.74	55.01 ± 18.47	0.4	0.8
Relational	58.97 ± 14.64	59.41 ± 12.07	59.11 ± 13.20	57.66 ± 12.13	1.6	0.2
Social	55.98 ± 16.19	58.80 ± 14.63	53.13 ± 16.21	57.61 ± 14.68	0.9	0.4
Environment	55.17 ± 14.02	56.85 ± 13.92	53.90 ± 18.12	55.34 ± 13.40	0.5	0.7
Tolerability	59.39 ± 18.10	56.44 ± 16.24	57.26 ± 17.59	56.97 ± 15.05	0.8	0.5
Core FertiQoL	50.37 ± 14.50	48.69 ± 18.78	38.28 ± 24.05	48.34 ± 16.31	0.7	0.6
Treatment FertiQoL	54.26 ± 12.02	51.60 ± 15.26	44.28 ± 21.81	52.40 ± 14.18	0.1	1.0
Total FertiQoL	55.04 ± 11.93	55.85 ± 11.14	52.78 ± 12.96	55.29 ± 10.69	0.5	0.7

## Data Availability

The datasets generated during and/or analyzed during the current study are available from the corresponding author on reasonable request.

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
