# Peer review of "Quality-of-Life Assessment of Women Undergoing In Vitro Fertilization in Kazakhstan"

_ijerph, 2022, doi:10.3390/ijerph192013568_

Round 1
Reviewer 1 Report
The manuscript “Quality of life assessment of women undergoing in vitro fertilization in Kazakhstan” is devoted to giving information about how to improve the quality of life of women suffering from infertility. The review is done concisely and soundly, although the presentation is somewhat dryish and formal. Anyway, this manuscript allowed me to recognize the problem of women suffering from infertility in the world as well as in Kazakhstan.
Some points have to be corrected.
Major points
1. There is not enough explanation about RK. Also, How many Kazakhs know Russian? (100%?) Or did only patients who knew Russian participate in the survey?
2. This tool has a limitation as described in the discussion. I agree with your opinion that men and men’s parents are other factors to influence FertiQoL of women undergoing infertility treatment. To develop a tool with axial sectional design, a follow-up study is needed to develop and apply it to verify its effectiveness. What various factors do we need to approach? For example, physical, psychological, and relational factors for each stage of IVF treatment. If possible, please describe it in detail.
Minor points
1. Line 20: Please change “suffering of” to “suffering from”.
2. Line 68: Please add “and” between “Iranian” and “German”.
3. Line 116: Please change “know” to “knows”.
4. Line 126: Please insert “are” before “divided”.
5. Line 132: Please delete space between “]” and “.” There are one space.
6. Line 134: Please change “was” to “were”
7. Line 191: Please insert “and” before “T5”
8. Line 210: Please change “affects” to “affect”.
9. Line 220: Please insert “,” after “In our study”.
10. Line 227: There are two spaces between “subsidezed” and “7000”.
11. Line 250: Please change “assess” to “assessment”.
12. n looks lonely in Figure 1. It will be better if Kazakhstan could be made into one line.
13. It is a little hard to read because there is no space between the table and the text. It needs to fix.
Reviewer 2 Report
While the study is relevant, the lower quality of the writing and data presentation hamper the readibility.
I restrict my comments to the content, considering a full review of the language is a conditio sine qua non for publication.
I would suggest to use the teams used in fertiqol (domain, score) and replace sphere, area,..
Please add the results and interpretation of the Total fertiqol score in the result section, as well as the analysis of the GDP, and the data on the infertility diagnosis and educational level.
Table 1; single should not be listed as infertility cause.
The anova test result should be reported as showing a difference between groups rather than between 2 specific groups
The results should maybe start with the Total fertiqol score rather than the domains.
The numbers of participants and response rate should be moved to the results section
All abbreviations should be spellen out at first use.
In the discussion, no new data should be presented. Listing the conclusion of this study, the results of other studies followed by the interpretation would again help to read and understand.
Reviewer 3 Report
This paper concerns the recently introduced “Fertility Quality of Life Questionnaire (Fer-19 tiQoL)”, which quickly became a gold standard for evaluation of the quality of life of patients suffering of infertility. The aim of this study was to determine the quality of life of Kazakhstani women coping with infertility problems by FertiQoL and assess the validity of questionnaire.
A limitation of the manuscript is the lack of a study concerning a quality of life assessment of men.
The bibliography should be updated with recent articles published on the subject.
The paper should be accepted after minor revisions.
Round 2
Reviewer 2 Report
Dear Authors,
In the initial review, I stated that I restricted my comments to the content, considering a full review of the language is a conditio sine qua non for publication. I consider this has not been done. To support you, I have now listed some of the most important language errors to be corrected;
Line 91 : remove "2." or add "1." I would also transform this into a proper sentence
Line 93: remove "for an online survey"
Line 94 : remove "set"
Line 97: remove "were assessed"
Line 110 :"rated on a scale from 0 to 4" - I suggest to add for what the items are rated?
Line 147 : remove "have"
Line 148 : change "have" to "had"
line 149: "most prevailed duration of infertility" => change to "most prevalent"
Line 158: change "no need to delete any items." to "there was no need to delete any items. "
Line 169: change "We investigated that respondents from rural areas had an emotional scale more than twice lower than in urban areas" to "When comparing the scores from respondents from rural areas with those from urban areas, we found that scores for the emotional scale were more than two times lower in respondents from rural areas'
Line 175 start the sentence with: When analysing differences in groups divided by income, Q4...
Line 180: Rephrase the sentence "The minimum duration of infertility up to two years showed high levels of quality of life compared to a longer period of infertility" to " when comparing respondents with a duration of infertility up to two years with those.."
Line 181 'The duration of infertility had .." this sentence is to be rewritten
Line 186: "There is no significant difference between education levels and cause of infertility" Please rephrase to indicate that there is no difference when comparing FertilQoL scores in groups divided by education level or reported cause of infertility.
Line 190: while I appreciate the paragrpah being moved, it need a subheading/background to be added so the reader can understand what is compared and why.
Line 207: add a full stop after questions, and rephrase the sentence starting at "other studies"
Line 211: rewrite "In Turkish study also no need to delete items"
Line 213: change "reflected results showing" to "showed"
Line 217: the sentence starting with "Even though" could be rephrased and clarified.
Line 219: This sentence should be split and clarified. The language of all sentence (line 220 tp 224) needs revising.
Line 225: rephrase. My suggestion "We also measured the differences of quality of life by place of residence and reported that respondents living in rural areas reported lower scores on (all/most) FertiQoL domains. This finding is consistent with findings of studies in Taiwan, Germany and Poland [12,23,25]".
Line 228 : change "worsen satisfaction" to "lower satisfaction" or decreased..
Line 230: change "51% of respondent" to "51% of the respondents"
Line 231: change "lower-than-average" to "below-average" as used in the results section. The sentence starting with "which in principle can" is not clear
Line 232: rephrase "The comparison by income level was carried out in Iran, which corresponds to our data" => "The comparison by income level was also done in a study from Iran, which reported data consistent with ours"
Line 235: rephrase the sentence starting with "This result finds out similarities"
Line 237 : Add "the" before "previous"
Line 240: "In a study by Koert et al., 2019, total scales of satisfaction with the quality of life were 240 compared across different countries" Maybe add the main conclusion, or remove this sentence
Line 241 : correct to "concluded"
Line 243 : Replace "it". The sentence is unclear.
Line 248: Start a new sentence after divorced.
Line 250: change "programs" to "treatments" or "cycles"
Line 253: Start a new sentence after "limitations"
Line 255: Also "the" cross-sectional design "limited"
Line 267: change "presented" to "present"
Line 278: change "infertility" to "these"
Line 285: change "European countries" to "scores reported in European countries"
Line 285: the sentence starting with "Questionnaire" does not make sense and needs rephrasing
